# Downlink Transmissions of UAV-RIS-Assisted Cell-Free Massive MIMO Systems: Location and Trajectory Optimization

**DOI:** 10.3390/s24134064

**Published:** 2024-06-22

**Authors:** Qi Zhang , Jie Zhao , Rongcheng Zhang , Longxiang Yang 

**Affiliations:** 1China Information Consulting & Design Institute Co., Ltd., Nanjing 210019, China; 2School of Communications and Information Engineering, Nanjing University of Posts and Telecommunications, Nanjing 210003, China; 1222014716@njupt.edu.cn (J.Z.); 1011020325@njupt.edu.cn (R.Z.); yanglx@njupt.edu.cn (L.Y.)

**Keywords:** CF-mMIMO, downlink sum rate, location and trajectory optimization, RIS, UAV

## Abstract

In this paper, we investigate a cell-free massive multiple-input multiple-output (CF-mMIMO) system with a reconfigurable intelligent surface (RIS) carried by an unmanned aerial vehicle (UAV), called the UAV-RIS. Compared with the RIS located on the ground, the UAV-RIS has a wider coverage that can reflect all signals from access points (APs) and user equipment (UE). By correlating the UAV location with the Rician *K*-factor, we derive a closed-form approximation of the UE achievable downlink rate. Based on this, we obtain the optimal UAV location and RIS phase shift that can maximize the UE sum rate through an alternating optimization method. Simulation results have verified the accuracy of the derived approximation and shown that the UE sum rate can be significantly improved with the obtained optimal UAV location and RIS phase shift. Moreover, we find that with a uniform UE distribution, the UAV-RIS should fly to the center of the system, while with an uneven UE distribution, the UAV-RIS should fly above the area where UEs are gathered. In addition, we also design the best trajectory for the UAV-RIS to fly from its initial location to the optimal destination while maintaining the maximum UE sum rate per time slot during the flight.

## 1. Introduction

Massive multiple-input multiple-output (mMIMO) is a promising technology in fifth-generation wireless communications (5G) [1]. It employs a large number of antennas for simultaneously serving only a few pieces of user equipment (UEs), leading to high spectral efficiency and energy-saving capability [2,3]. However, severe inter-cell interference is the main bottleneck that hinders its performance improvement. To address this issue, cell-free massive MIMO (CF-mMIMO) has been proposed [4]. In CF-mMIMO, the collaborative operation of numerous distributed access points (APs) that connect the central processing unit (CPU) through the backhaul is used to serve all user UEs simultaneously. CF-mMIMO eliminates the cell boundary and can effectively address the inter-cell interference prevalent in conventional cellular networks [5,6]. As there is no handover among cells in CF-mMIMO, it can provide uniform service for UEs [7]. In addition, CF-mMIMO exploits the channel-hardening property when the number of APs is large enough to offer significant performance with simple signal processing [8]. Furthermore, as UEs are located in close proximity to APs, it yields notable improvements in the UE spectral efficiency due to the rich macro-diversity gain, as well as the short transmission delay [9]. Given this, CF-mMIMO has emerged as a key technology in next-generation wireless communications. However, the practical application of CF-mMIMO faces certain challenges, primarily associated with the high hardware costs stemming from large numbers of APs and backhaul fibers. These cost considerations impose practical limitations on the widespread deployment of CF-mMIMO systems.

The reconfigurable intelligent surface (RIS), which is a planar array comprising numerous low-cost passive reflective components, has emerged as a new popular technique [10,11,12]. It is positioned between transmitters and receivers to provide an additional channel environment and enhance communication quality [13]. Each element of the RIS possesses the capability to independently manipulate the phase and amplitude of incident signals, enabling easy deployment for communication system capacity enhancement without requiring any power amplifiers [14,15]. It can offer significant advantages in terms of reducing energy consumption and hardware costs, as well as flexible installation. As a result, the RIS emerges as a crucial facilitator of environmentally friendly communication systems of the future [16,17]. Applying an RIS in CF-mMIMO can effectively reduce the hardware costs of APs and backhauls while maintaining a satisfactory spectral efficiency [18].

Most previous works integrating an RIS into CF-mMIMO put the RIS on the ground, which cannot cover all UEs due to the one-side reflection feature of the RIS. An unmanned aerial vehicle (UAV)-mounted RIS offers a solution to overcome this limitation. UAVs have gained great interest recently due to their high mobility, easy deployment and reliable line-of-sight (LoS) transmissions. The integration of RISs and UAVs has been explored in a number of studies to improve the performance of air-to-ground networks. In [19], the authors analyzed the RIS-assisted UAV in multi-user communication and jointly optimized the phase shift and UAV trajectory. The authors in [20] considered that the UAV carrying an RIS serves as a mobile relay to implement the classic three-node cooperative communication model, and the UAV trajectory was optimized to improve the cooperative performance. However, these works were performed in traditional cellular systems. In [21], the authors studied the combination of RIS and UAV CF-mMIMO systems, but they assumed that there exists only the reflective link and a single UE. In [22], the RIS was mounted on the UAV to serve users in different groups, and deep learning was used to optimize the UAV location and RIS phase shift. However, this study did not correlate the channel model with UAV locations.

In this paper, we study a CF-mMIMO system with multiple UEs, which is assisted by a UAV-mounted RIS, abbreviated as the UAV-RIS. Both the direct and reflective links between APs and UEs are considered, and the reflection channel through the UAV-RIS is modeled in association with the UAV altitude. A closed-form approximation of the UE achievable downlink rate is derived. On top of this, we obtain the optimal UAV location and RIS phase shift that can maximize the UE sum rate with an alternating optimization method. Simulations have shown that the UE sum rate can be significantly improved with the obtained UAV location and RIS phase shift. Moreover, we also optimize the UAV trajectory to maximize the UE sum rate during the flight from the UAV’s initial location to the optimal location.

*Notation*: Within this paper, a vector is represented by a bold lower-case letter, whereas a matrix is indicated by an upper-case letter. $a^{*}$ and $a^{H}$ denote the conjugate of and the conjugate transpose of a, respectively. Let E{·} denote the expectation operator, and $I_{L}$ denote the identity matrix with a size of L×L.

## 2. System Model

Consider a CF-mMIMO system with *M* APs and *K* UEs. Each AP connects the CPU through a fiber backhaul. APs and UEs both have a single antenna (as APs cooperate through the backhaul, we can combine them into a “virtual MIMO”). A RIS mounted on the UAV, called a UAV-RIS, is used in this system, as shown in Figure 1. The location coordinate of the UAV-RIS is $\left(x_{0},y_{0},h_{0}\right)$. Each AP and UE are randomly distributed in the system, and the area of the system is denoted by *S*.

Due to the fact that APs and UEs are located on the ground, they experience a significant presence of buildings and scatters. Therefore, we model the channel between APs and UEs as Rayleigh fading. We formulate the direct channel between the *k*-th UE and the *m*-th AP as [4]
(1)$$\begin{array}{c} g_{mk}=h_{mk}\sqrt{\beta_{mk}}, \end{array}$$
where $h_{mk}$ is the fast fading coefficient between the *k*-th UE and the *m*-th AP, which is a complex Gaussian variable with a zero mean and unit variance. That is, $h_{mk}∼\mathrm{CN}\left(0,1\right)$. $\beta_{mk}$ is the large-scale fading between the *k*-th UE and the *m*-th AP, and it includes geometric attenuation. As the RIS is mounted on the UAV, the absence of obstacles in the aerial transmission makes the LoS become dominant. Therefore, channels between APs and the RIS, as well as channels between the RIS and UEs, will contain LoS components. We use Rician fading [23], which encompasses a dominant LoS component along with a Rayleigh-distributed random component, to model them. The channel between the *m*-th AP and the RIS is expressed as
(2)$$\begin{array}{c} g_{m}=\sqrt{\varepsilon_{m}}\left(\sqrt{\frac{K_{1,m}}{K_{1,m}+1}}{\bar{g}}_{m}+\sqrt{\frac{1}{K_{1,m}+1}}{\hat{g}}_{m}\right), \end{array}$$
and the channel between the *k*-th UE and the RIS is described by
(3)$$\begin{array}{c} h_{k}=\sqrt{\zeta_{k}}\left(\sqrt{\frac{K_{2,k}}{K_{2,k}+1}}{\bar{h}}_{k}+\sqrt{\frac{1}{K_{2,k}+1}}{\hat{h}}_{k}\right), \end{array}$$
where $\varepsilon_{m}$ and $\zeta_{k}$ represent the large-scale fading, and $K_{1,m}$ and $K_{2,k}$ are the Rician *K*-factors. The value of the Rician *K*-factor is related to the UAV location. ${\bar{g}}_{m}\in C^{L\times 1}$ and ${\bar{h}}_{k}\in C^{1\times L}$ are the LoS components, and ${\hat{g}}_{m}\in C^{L\times 1}$ and ${\hat{g}}_{k}\in C^{1\times L}$ are the non-LoS (NLoS) components [24]. The elements of ${\bar{g}}_{m}$ and ${\bar{h}}_{k}$ are
(4)$$\begin{array}{c} {\left[{\bar{g}}_{m}\right]}_{l}=e^{j\left(l-1\right)\pi sin\left(\theta_{m}^{a}\right)}, \end{array}$$
and
(5)$${\left[{\bar{h}}_{k}\right]}_{l}=e^{j\left(l-1\right)\pi sin\left(\theta_{k}^{u}\right)},$$
respectively, where $\theta_{m}^{a}$ represents the angle of arrival (AoA) from the *m*-th AP to the RIS, and $\theta_{k}^{u}$ represents the angle of departure (AoD) from the RIS to the *k*-th UE [25].

## 3. Downlink Transmissions

In this section, the ergodic downlink rate of the UE is derived, and we obtain the optimal UAV location and RIS phase shift that maximize the downlink sum rate of all UEs. We assume that the channel state information (CSI) between APs, UEs and UAV-RIS can be obtained perfectly. Let $s_{k}$ denote the symbol intended for the *k*-th UE, satisfying $E\{|s_{k}{|}^{2}\}=1$. The transmitted signal from the *m*-th AP is given by
(6)$$\begin{array}{c} x_{m}=\sum_{k=1}^{K}\sqrt{\rho_{d}}{\left(g_{mk}+g_{m}^{H}\Phi h_{k}\right)}^{*}s_{k}, \end{array}$$
where $\Phi =\mathrm{diag}\{\phi_{1},\phi_{2},\cdot \cdot \cdot ,\phi_{L}\}$ denotes the diagonal phase-shifting matrix at the RIS with $\phi_{1},\ldots ,\phi_{L}$ as the diagonal elements, and $\phi_{l}=e^{j\theta_{l}},\theta_{l}\in \left[0,2\pi \right)$. $g_{mk}+g_{m}^{H}\Phi h_{k}$ denotes the cascaded link between the AP and the UE, and $\rho_{d}$ denotes the AP transmit power.

The signal received by the *k*-th UE is expressed as
(7)$$\begin{array}{cc} y_{k} & =\sum_{m=1}^{M}\left(g_{mk}+g_{m}^{H}\Phi h_{k}\right)x_{m}+n_{k}, \end{array}$$
where $n_{k}$ is the additive white Gaussian noise at the *k*-th UE following CN(0,1) [4]. Then, (Equation 7) can be rewritten as
(8)$$\begin{array}{c} y_{k}=\;\;\sum_{m=1}^{M}\;\;\sqrt{\rho_{d}}t_{mk}t_{mk}^{*}s_{k}\;+\;\;\sum_{m=1}^{M}\sum_{j=1,j\neq k}^{K}\;\;\;\;\sqrt{\rho_{d}}t_{mk}t_{mj}^{*}s_{j}+n_{k}, \end{array}$$
where $t_{mk}=g_{mk}+g_{m}^{H}\Phi h_{k}$. As the LoS component changes slowly, we obtain the ergodic achievable rate of the UE by taking the average over all states of the NLoS component. Then, the ergodic achievable rate of the *k*-th UE is
(9)$$R_{k}=E\left\{\log_{2}\left(1+\frac{{\left|\sum_{m=1}^{M}\sqrt{\rho_{d}}{\left|t_{mk}\right|}^{2}\right|}^{2}}{\sum_{j=1,j\neq k}^{K}{\left|\sum_{m=1}^{M}\sqrt{\rho_{d}}t_{mk}t_{mj}^{*}\right|}^{2}+1}\right)\right\}.$$

The closed-form approximation of (Equation 9) is given by the following theorem.

**Theorem 1.** 
*A closed-form approximation for the achievable downlink rate of the k-th UE is given by*

(10)
$$\begin{array}{cc} \; & R_{k}\approx \tilde{R_{k}}=\\ \; & \log_{2}\left(1+\frac{\rho_{d}{\left(\sum_{m=1}^{M}F\left(m,k\right)\right)}^{2}+\lambda_{k}}{\sum_{j=1,j\neq k}^{K}\left(\sum_{m=1}^{M}\rho_{d}F\left(m,k\right)F\left(m,j\right)+\sum_{m=1}^{M}\sum_{n=1,n\neq m}^{M}\rho_{d}A\left(m,n,k\right)\right)+1}\right). \end{array}$$

*where*

(11)
$$\begin{array}{cc} A\left(m,n,k\right) & ≜T\left(m,k\right)T^{*}\left(m,j\right)T^{*}\left(n,k\right)T\left(n,j\right) \end{array}$$


(12)
$$\begin{array}{cc} F(m,j & ≜\beta_{mj}+r_{mj}^{2}\;\left({\left|\chi_{1,mk}\right|}^{2}+\kappa_{mk}\right) \end{array}$$


(13)
$$\begin{array}{cc} T(m,k) & ≜r_{mk}\chi_{1,mk} \end{array}$$


(14)
$$\begin{array}{cc} \lambda_{k} & ≜\sum_{m=1}^{M}\rho_{d}\left(\beta_{mk}^{2}+2\beta_{mk}r_{mk}^{2}\left(\right|\chi_{1,mk}{|}^{2}+\kappa_{mk})+r_{mk}^{4}V_{mk}\right), \end{array}$$

*with $\chi_{1,mk}≜\sqrt{K_{1,m}K_{2,k}}{\bar{g}}_{m}^{H}\Phi {\bar{h}}_{k}$, $r_{mk}≜\sqrt{\varepsilon_{m}\zeta_{k}/\left(K_{1,m}+1\right)\left(K_{2,k}+1\right)}$, $\kappa_{mk}≜K_{1,m}+K_{2,k}+1$, and*

(15)
$$\begin{array}{cc} \; & V_{mk}≜K_{1,m}^{2}L^{2}+K_{2,k}^{2}L^{2}+L^{2}+2L+2\kappa_{mk}{\left|\chi_{1,mk}\right|}^{2}\\ \; & \;\;\;\;\;\;\;\;\;\;\;+2K_{1,m}K_{2,k}L^{2}+2K_{1,m}L^{2}+2K_{2,k}L^{2}+4K_{1,m}L+4K_{2,k}L. \end{array}$$



**Proof.** See Appendix A.    □

The accuracy of (Equation 10) will be verified in Section 5 with simulations. From Theorem 1, it is evident that the downlink rate of the UE is contingent upon the distribution of APs and UEs. Moreover, it is also affected by the location and phase shift of the UAV-RIS. We assume that APs and UEs are fixed, and we can adjust the location and phase shift of the UAV-RIS to obtain a better rate. Therefore, the subsequent section will concentrate on determining the optimal location and phase shift of the UAV-RIS, aiming to maximize the UE downlink sum rate.

## 4. UAV-RIS Optimization

### 4.1. Location and Phase-Shift Optimization

In this section, our main goal is to identify the optimal phase shift of the RIS and to determine the optimal location of the UAV. This optimization aims to maximize the downlink sum rate of UEs. To accomplish this, we will address the following problem:$$\begin{array}{cc} \left(P1\right)\;\;\;\max_{\Phi ,x_{0},y_{0},h_{0}}\;\; & f\left(\Phi ,x_{0},y_{0},h_{0}\right)=\sum_{k=1}^{K}{\tilde{R}}_{k} \end{array}$$(16)$$\begin{array}{cc} s.t.\;\; & 0\leq \theta_{l}<2\pi ,\;\forall l. \end{array}$$(17)$$\begin{array}{cc} \;\; & (x_{0},y_{0},h_{0})\in S. \end{array}$$

Problem (P1) has multiple variables and is non-convex. Hence, we use an alternating method to divide it into two subproblems: phase-shift optimization and location optimization [5]. Then, each subproblem is iteratively solved until the final results converge.

#### 4.1.1. Fix $\left(x_{0}^{★},y_{0}^{★},h_{0}^{★}\right)$ and Solve $\Phi^{opt}$

First, we fix the UAV location at $\left(x_{0}^{★},y_{0}^{★},h_{0}^{★}\right)$ and optimize the RIS phase shift. For notation simplification, we define the diagonal vector of Φ as
(18)$$\Theta ≜[e^{j\theta_{1}},\cdots ,e^{j\theta_{L}}].$$

Then, we need to solve the following problem:(19)$$\begin{array}{cc} \left(P2\right)\;\;\;\max_{\Theta }\;\; & f\left(\Theta ,x_{0}^{★},y_{0}^{★},h_{0}^{★}\right)=\sum_{k=1}^{K}{\tilde{R}}_{k} \end{array}$$

In order to solve the phase shift more conveniently, we make the phase shift explicit in the objective function, and ${\tilde{R}}_{k}$ is rewritten as
(20)$$\begin{array}{c} {\tilde{R}}_{k}=\log_{2}\left(1+\frac{\Theta P_{k}\Theta^{H}+{\left|\Theta Z_{k,k}\Theta^{H}\right|}^{2}}{\Theta B_{k}\Theta^{H}+\sum_{j=1,j\neq k}{\left|\Theta Z_{k,j}\Theta^{H}\right|}^{2}}\right), \end{array}$$
where
(21)$$\begin{array}{cc} B_{k} & ≜\frac{1}{L}I_{L}I_{k}\;+\;\rho_{d}\;\;\;\sum_{j=1,j\neq k}^{K}\sum_{m=1}^{M}\left(\Xi_{mj}\mu_{mk}+\Xi_{mk}\mu_{mj}\right), \end{array}$$
(22)$$\begin{array}{cc} P_{k} & ≜\frac{1}{L}I_{L}\left[\sum_{m=1}^{M}\rho_{d}\left(\beta_{mk}^{2}+r_{mk}^{4}I_{mk}+2r_{mk}^{2}\beta_{mk}\kappa_{mk}\right)+\sum_{m=1}^{M}\sum_{n=1}^{M}\rho_{d}\mu_{mk}\mu_{nk}\right]\\ \; & \;\;\;\;+2\rho_{d}\sum_{m=1}^{M}\beta_{mk}\Xi_{mk}+\rho_{d}\sum_{m=1}^{M}\sum_{n=1}^{M}\left(\Xi_{nk}\mu_{mk}+\Xi_{mk}\mu_{nk}\right), \end{array}$$
(23)$$\begin{array}{cc} Z_{k,j} & ≜\sum_{m=1}^{M}\sqrt{\rho_{d}}\Lambda_{mk}\Lambda_{mj}^{H}, \end{array}$$
with $\mu_{mk}≜\beta_{mk}+r_{mk}^{2}\kappa_{mk},\;\Xi_{mk}≜\Lambda_{mk}\Lambda_{mk}^{H},\;I_{k}≜\rho_{d}\;\;\;\sum_{j=1,j\neq k}^{K}\sum_{m=1}^{M}\mu_{mk}\mu_{mj}$, and $\Lambda_{mk}≜r_{mk}$$\mathrm{diag}\left({\bar{g}}_{m}{\bar{h}}_{k}^{H}\right)\sqrt{K_{1,m}K_{2,k}}.$

As problem (P2) is still non-convex, we use the gradient descent method to optimize the phase shift. The gradient vector of $f(\Theta ,x_{0},y_{0},h_{0})$ with respect to Θ is denoted  by
(24)$$\begin{array}{c} p=\sum_{k=1}^{K}\frac{1}{\ln2}{\left(1+\frac{\Theta P_{k}\Theta^{H}+{\left|\Theta Z_{k,k}\Theta^{H}\right|}^{2}}{\Theta B_{k}\Theta^{H}+\;\;\;\sum_{j=1,j\neq k}^{K}\;\;{\left|\Theta Z_{k,j}\Theta^{H}\right|}^{2}}\right)}^{-1}\;\;\;\;\;\;Q\left(\Theta \right), \end{array}$$
where
(25)$$\begin{array}{c} Q\left(\Theta \right)=\frac{Q_{1}\left(\Theta \right)-Q_{2}\left(\Theta \right)}{{\left(\Theta B_{k}\Theta^{H}+\sum_{j=1,j\neq k}^{K}{\left|\Theta Z_{k,j}\Theta^{H}\right|}^{2}\right)}^{2}}, \end{array}$$
with
(26)$$\begin{array}{cc} Q_{1}\left(\Theta \right) & =\left(P_{k}^{T}\Theta^{T}+2\Theta Z_{k,k}\Theta^{H}Z_{k,k}^{T}\Theta^{T}\right)\left(\;\Theta B_{k}\Theta^{H}+\;\;\sum_{j=1,j\neq k}^{K}{\left|\Theta Z_{k,j}\Theta^{H}\right|}^{2}\right), \end{array}$$
(27)$$\begin{array}{cc} Q_{2}\left(\Theta \right) & =\left(\Theta P_{k}\Theta^{H}+{\left|\Theta Z_{k,k}\Theta^{H}\right|}^{2}\right)\left(B_{k}^{T}\Theta^{T}+\sum_{j=1,j\neq k}^{K}2\Theta Z_{k,j}\Theta^{H}Z_{k,j}^{T}\Theta^{T}\right). \end{array}$$

In each iteration, the next phase-shift vector is computed based on the current gradient vector result. This iterative process continues until the UE sum rate reaches a convergent state.

#### 4.1.2. Fix $\Phi^{★}$ and Solve $\left(x_{0}^{opt},y_{0}^{opt},h_{0}^{opt}\right)$

Secondly, we fix the phase shift as $\Phi^{★}$ and optimize the UAV-RIS location. That is, we need to solve the following problem:(28)$$\begin{array}{cc} \left(P3\right)\;\;\;\max_{x_{0},y_{0},h_{0}}\;\; & f(\Phi^{★},x_{0},y_{0},h_{0})\\ s.t.\;\; & (x_{0},y_{0},h_{0})\in S. \end{array}$$
Constraint (Equation 28) requires that the UAV-RIS remain within the designated *S*. To solve (P3), we still use the iterative gradient descent method. In each iteration, we need to determine whether the location exceeds the area constraint. The gradient vector of the UAV-RIS location is given by
(29)$$\begin{array}{cc} q= & {\left[\frac{\partial f(\Phi ,x_{0},y_{0},h_{0})}{\partial x_{0}},\frac{\partial f(\Phi ,x_{0},y_{0},h_{0})}{\partial y_{0}},\frac{\partial f(\Phi ,x_{0},y_{0},h_{0})}{\partial h_{0}}\right]}^{T}. \end{array}$$

The closed-form expression of q is omitted due to space limitations. During the iteration, in the event that the UAV-RIS flies outside the defined area, it is essential to search for a feasible solution in the opposite direction of the gradient and continue iterating.

#### 4.1.3. Complete Algorithm

With the above two methods, the optimization process alternates between updating the phase shift Θ and the location $\left(x_{0},y_{0},h_{0}\right)$ until the UE sum rate converges. Note that, except for the initialization, the fixed values in problems (P2) and (P3) are obtained from the optimal outputs of each other. Algorithm 1 provides a concise summary of the complete alternating optimization process.
**Algorithm 1** Alternating Optimization Algorithm for (P1).**Initialization**: $\Theta^{0}=\Theta^{ran}$, $\omega^{0}={\left(x_{0}^{ran},y_{0}^{ran},h_{0}^{ran}\right)}^{T}$, search step $\upsilon_{1}=0.01$, $\upsilon_{2}=0.1$, $\upsilon_{3}=1$, error $\epsilon =10^{-6}$, and i=0, j=0, e=0.1: **Repeat**2:  **Repeat**3:    Obtain the gradient vector $p^{i}$ based on (Equation 24).4:    Obtain ${\hat{\Theta }}^{i+1}=\Theta^{i}+\upsilon_{1}p^{i}$ and $\Theta^{i+1}=e^{j\arg({\hat{\Theta }}^{i+1})}$.5:    Obtain the sum rate $f(\Theta^{i+1})$.6:    Update i=i+1.7:  **Until** $|f\left(\Theta^{i+1}\right)-f\left(\Theta^{i}\right)|<\epsilon$.8:   $\Theta^{e}=\Theta^{i+1}$.9:  **Repeat**10:   Obtain the gradient vector $q^{j}$ based on (Equation 29).11:   Obtain $\omega^{j+1}=\omega^{j}+\upsilon_{2}q^{j}$.12:   Determine whether the location exceeds the area; if it       does, let $\omega^{j+1}=\omega^{j}-\upsilon_{3}q^{l}$, and go to 14.13:   Obtain the sum rate $f(\omega^{j+1})$.14:   Update j=j+1.     **Until** $|f\left(\omega^{i+1}\right)-f\left(\omega^{i}\right)|<\epsilon$.15:  $\omega^{opt}=\omega^{l+1}$.16: Update e=e+1, $\Theta^{e}=\Theta^{e-1},$$\omega^{e}=\omega^{opt}$17: **Until** $|f\left(\Theta^{e},\omega^{e}\right)-f\left(\Theta^{e-1},\omega^{e-1}\right)|<\epsilon ,$18: $\Theta^{opt}=\Theta^{e},\left(x_{0}^{opt},y_{0}^{opt},h_{0}^{opt}\right)=\omega^{e}.$**Output**: $\Theta^{opt}$ and $(x_{0}^{opt},y_{0}^{opt},h_{0}^{opt}).$

### 4.2. Trajectory Optimization

After obtaining the optimal UAV-RIS location, we aim to find the best trajectory for the UAV-RIS flight from the initial location to the optimal destination. Assume that the whole flight costs *T* time slots. Let $R_{k}\left[i\right]$, (x[i],y[i],h[i]) and Θ[i] denote the achievable rate, location coordinate and phase shift at the *i*-th time slot, respectively. Define the distance traveled by the UAV-RIS from the *i*-th time slot to the i+1-th time slot as
(30)$$\begin{array}{c} D\left[i,i+1\right]=∥\left(x[i+1],y[i+1],h[i+1]\right)-\left(x[i],y[i],h[i]\right)∥, \end{array}$$
and the distance between the UAV-RIS location at the *i*-th time slot and its optimal location as
(31)$$\begin{array}{c} \tilde{D}\left[i,opt\right]=∥\left(x[i],y[i],h[i]\right)-\left(x_{0}^{opt},y_{0}^{opt},h_{0}^{opt}\right)∥. \end{array}$$

Then, the trajectory optimization problem is formulated as
$$\begin{array}{cc} \left(P4\right)\max_{T,\;(x[i],y[i],h[i]),\;\Theta [i]}\;\;\; & \frac{1}{T}\sum_{i=1}^{T}\sum_{k=1}^{K}{\tilde{R}}_{k}\left[i\right] \end{array}$$
(32)$$\begin{array}{cc} s.t.\;\;\;\;\;\;\;\;\; & D\left[i,i+1\right]\leq D_{max}, \end{array}$$
(33)$$\begin{array}{cc} \; & \tilde{D}\left[T,opt\right]<D_{max}, \end{array}$$
(34)$$\begin{array}{cc} \; & \left(x[i],y[i],h[i]\right)\in S, \end{array}$$
(35)$$\begin{array}{cc} \; & T\geq 1. \end{array}$$
where $D_{max}$ represents the maximum distance that the UAV-RIS can move in one time slot. The objective function is the average sum rate over all time slots. Constraint (Equation 33) means that, in the final time slot, the UAV-RIS should fly to near the optimal location. Constraint (Equation 34) means that the UAV-RIS cannot fly outside the system area. This problem can be decomposed into two parts to solve: one is solving the time slot, and the other is solving the UAV-RIS trajectory as well as the phase shift.

As *T* is an integer, we fix it and then solve the optimal trajectory and phase shift. That is, problem (P4) becomes
$$\begin{array}{cc} \left(P5\right)\max_{\left(x[i],y[i],h[i]),\;\Theta [i\right]}\;\;\; & \frac{1}{T}\sum_{i=1}^{T}\sum_{k=1}^{K}{\tilde{R}}_{k}\left[i\right] \end{array}$$
(36)$$\begin{array}{cc} s.t.\;\;\;\;\;\;\;\;\; & D\left[i,i+1\right]\leq D_{max}, \end{array}$$
(37)$$\begin{array}{cc} \; & \tilde{D}\left[T,opt\right]<D_{max}, \end{array}$$
(38)$$\begin{array}{cc} \; & \left(x[i],y[i],h[i]\right)\in S. \end{array}$$

Using the same method used to solve (P1), we can solve (P5) through an alternatively iterative method that optimizes the UAV-RIS location in each time slot separately. Note that when the distance difference between the locations in two adjacent time slots is greater than $D_{max}$, the location will be re-updated until the distance between the current location and the destination location is less than the threshold.

We exhaustively search for *T* and solve problem (P5) with each fixed *T*. We find that as *T* keeps growing, the UAV-RIS approaches the final location, and the average sum rate gradually stabilizes. Hence, we can obtain the optimal $T^{opt}$ at the convergence point. The change in the average sum rate with respect to *T* is shown in the numerical results.

## 5. Numerical Results

In this section, we begin by confirming the precision of the derived downlink rate approximation as mentioned in Theorem 1 and subsequently examine the efficiency of our optimized outcomes.

### 5.1. Parameters and Setup

In our system, *M* APs and *K* UEs are uniformly and randomly distributed over a square area with a size of 400×400 m^2^. The Ricean *K*-factor in the channel model is given by [26]
(39)$$\begin{array}{c} K_{1,m}=\frac{p_{LoS}\left(d_{1,m}\right)}{1+p_{LoS}\left(d_{1,m}\right)}, \end{array}$$
where $d_{1,m}$ represents the distance (in meters) between the *m*-th AP and the UAV-RIS, and $p_{LoS}\left(d_{1,m}\right)$ denotes the LoS probability contingent on the distance $d_{1,m}$. We utilize the model in 3GPP to calculate $p_{LoS}$ [26] as follows:(40)$$\begin{array}{c} p_{LoS}\left(d_{1,m}\right)=\min\left(\frac{18}{d_{1,m}},1\right)\left(1-e^{-\frac{d_{1,m}}{63}}\right)+e^{-\frac{d_{1,m}}{63}}. \end{array}$$

Similarly, $K_{2,k}$ is defined in the same way as (Equation 39), where $d_{1,m}$ is replaced with $d_{2,k}$. $d_{2,k}$ denotes the distance (in meters) between the *k*-th UE and the UAV-RIS. We define the large-scale fading coefficients as $\beta_{mk}=1/d_{mk}^{\alpha },\;\varepsilon_{m}=1/d_{1,m}^{\alpha },\;\zeta_{k}=d_{2,k}^{-\alpha }$, where α is the path loss exponent, and $d_{mk}$ represents the distance between the *m*-th AP and the *k*-th UE. The path loss exponent α is commonly set to 2.

### 5.2. Performance Analysis

Figure 2 illustrates a comparison between the simulated downlink achievable sum rate according to (Equation 9) and its analytical approximation in (Equation 10). The phase shift of the RIS is given randomly. Clearly, we can see a precise agreement between the simulation and our approximation results, which verifies the accuracy of our analytical derivations. We can also find that as *M* grows, the sum rate increases considerably, since APs can serve all UEs in CF-mMIMO, and more APs enable more effective signals. Due to the close agreement between the simulation and analysis, we use the latter for the following investigation.

In Figure 3, we show the effectiveness of our optimal results. The downlink sum rates with a UAV-RIS having a jointly optimal phase shift and location, having only an optimal phase shift, and having no optimization are compared. The optimization result is from Algorithm 1. The non-optimal phase shift and location are given randomly. We can see that after optimizing the UAV-RIS phase shift, there is a significant improvement in the downlink sum rate. This improvement becomes more pronounced after jointly optimizing the phase shift and location of the UAV-RIS. Moreover, as the number of RIS components grows, the sum rate also has a notable increment, since more RIS components provide more reflective paths to collect effective signals.

In order to illustrate the trajectory optimization problem, we show the average sum rate over time slots with respect to the flight time slot in Figure 4. For each fixed time slot, we solve the UAV-RIS location and phase shift, as well as the corresponding maximum downlink sum rate according to problem (P5). The average sum rate over time slots in Figure 4 is obtained by dividing the downlink sum rate by the time slot it requires. In Figure 4, we can see that as the time slot number increases, the average sum rate gradually increases and finally tends to be stable. Hence, for energy saving, we should choose the smallest time slot with the peak value of the average sum rate as the optimal one. In Figure 4, the optimal flight time slot is $T^{opt}=25$.

With the optimal time slot obtained from Figure 4, the UAV-RIS trajectory can be obtained by solving the optimal UAV-RIS location in each time slot within $\left[1,T^{opt}\right]$, according to problem (P5). Then, the UAV-RIS trajectory is shown in Figure 5 with a uniform UE distribution and in Figure 6 with an uneven UE distribution. The initial location of the UAV-RIS is (0,0,60). We can see that when UEs are distributed uniformly, the UAV-RIS should fly to the center of the system to maximize the UE sum rate. However, when UEs are gathered in a limited area, the UAV-RIS should fly above the region where UEs are gathered. It can also be noted that the trajectory of the UAV-RIS is not simply a straight line from the starting point to the destination. Instead, it curves through areas with higher average sum rates to maximize the average sum rate of all UEs.

## 6. Conclusions

In this paper, we have developed a framework for the CF-mMIMO system with a UAV-RIS. A closed-form approximation of the UE achievable downlink rate has been derived, and its accuracy has been verified via simulations. On top of this, the optimal phase shift and UAV-RIS location that maximize the downlink sum rate of all UEs have been obtained through an alternating optimization method. We have shown that with our optimal UAV-RIS deployment results, the downlink sum rate can be significantly improved. It is found that, with a uniform UE distribution, the UAV-RIS should fly to the center of the system, while with an uneven UE distribution, the UAV-RIS should fly above the area where UEs are gathered. Moreover, we have also explored the optimal UAV-RIS trajectory from its starting point to the optimal location, which can keep the downlink sum rate in each time slot maximized. The assumption of perfect CSI is a limitation of this work, restricting its practical application. In the future, we will extend this paper to the case with imperfect CSI.

## Figures and Tables

**Figure 1 sensors-24-04064-f001:**
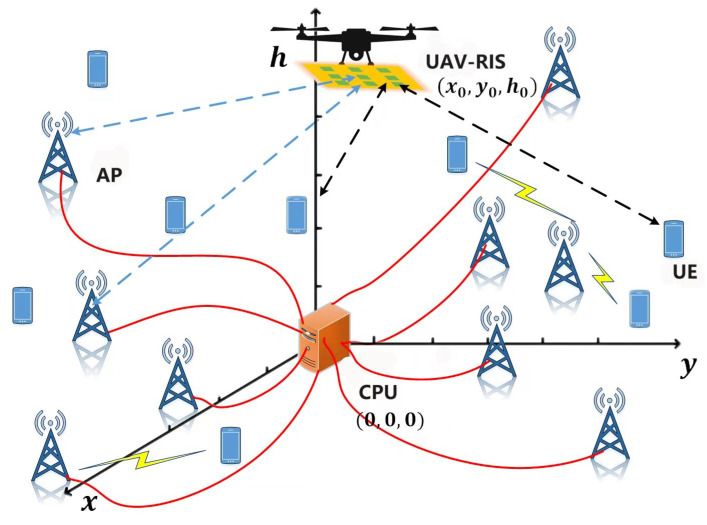
The CF-mMIMO system with a UAV-RIS.

**Figure 2 sensors-24-04064-f002:**
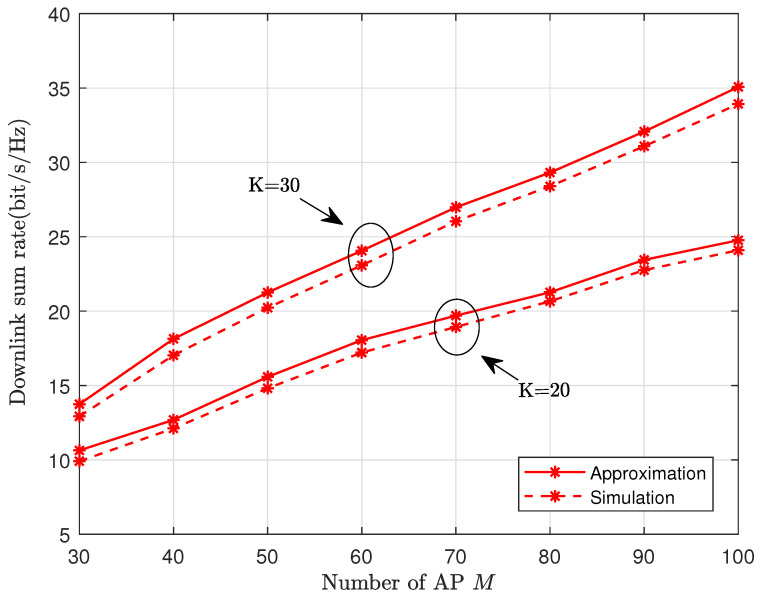
The downlink sum rate vs. the AP number, where L=64, and the location of the UAV-RIS is (0,0,60) in meters.

**Figure 3 sensors-24-04064-f003:**
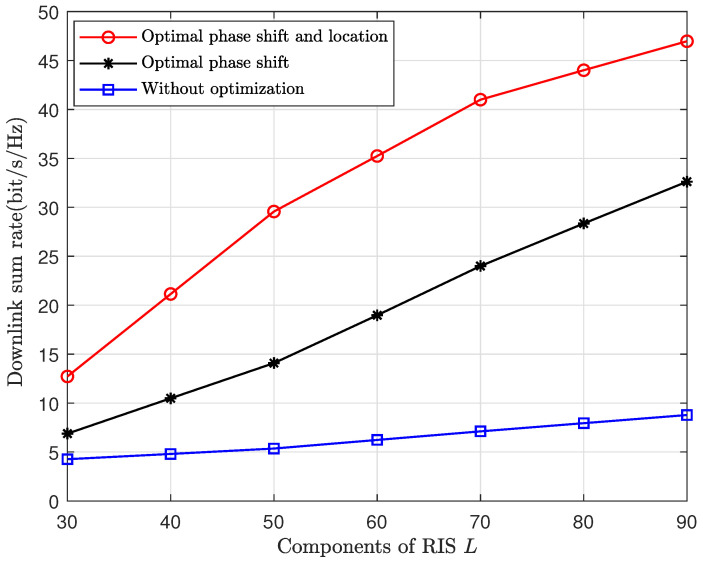
The downlink sum rate vs. the number of RIS components, where M=30 and K=20.

**Figure 4 sensors-24-04064-f004:**
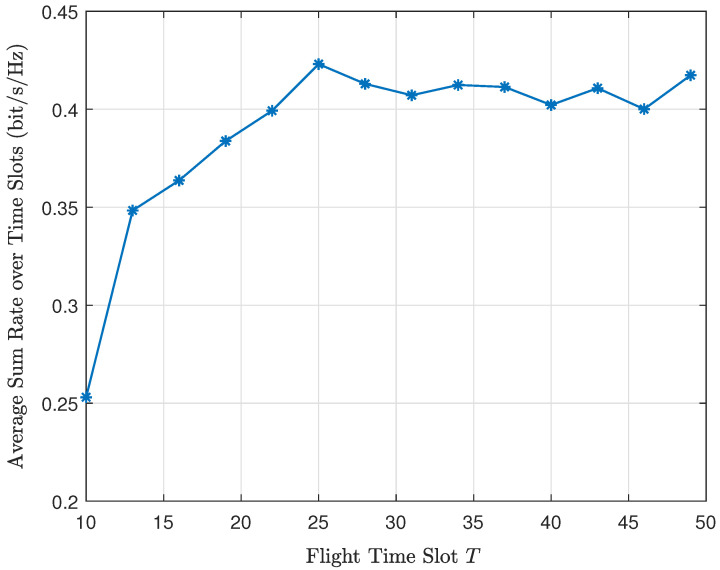
The average sum rate vs. the flight time slot, where M=6, K=4, L=50, and the initial location of the UAV-RIS is (0,0,60) in meters.

**Figure 5 sensors-24-04064-f005:**
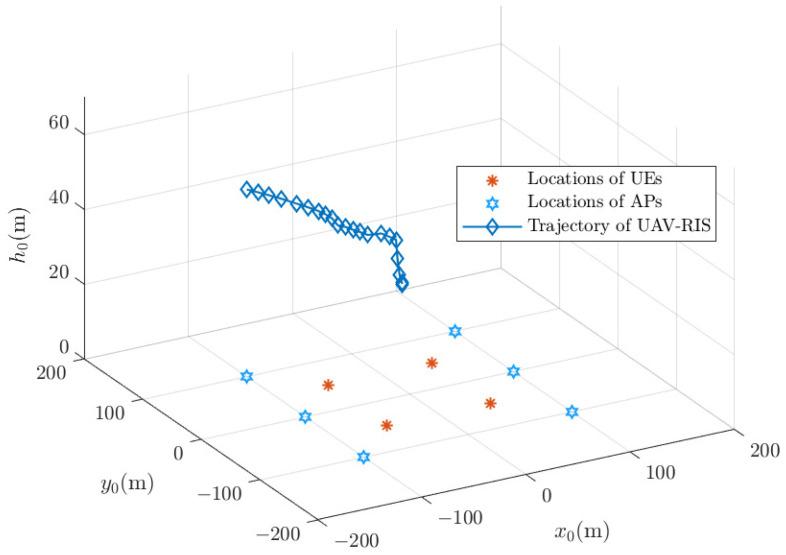
The optimal trajectory of the UAV-RIS with a uniform UE distribution.

**Figure 6 sensors-24-04064-f006:**
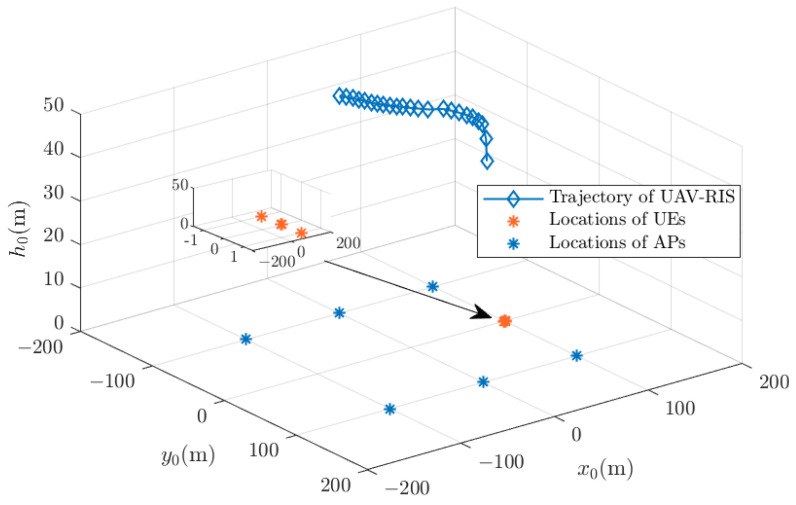
The optimal trajectory of the UAV-RIS with an uneven UE distribution.

## Data Availability

Data are contained within the article.

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
