# Peer review of "Downlink Transmissions of UAV-RIS-Assisted Cell-Free Massive MIMO Systems: Location and Trajectory Optimization"

_sensors, 2024, doi:10.3390/s24134064_

Round 1
Reviewer 1 Report
Comments and Suggestions for Authors
This paper investigated an optimal problem of the localization and trajectory of UAV-RIS for a Cell-Free Massive MIMO system to maximize the sum rates of all users. The experimental results showed the validation of the optimal scheme. However, there are some following problems that need to be addressed:
1) The application background of this paper is Cell-Free Massive MIMO systems, which not only require a large number of AP points, but also require a large-scale antenna array at both ends of the transmitter and receiver. However, the APs and users in this paper only consider the case of a single antenna, which obviously simplifies the scene design and does not match the application background. It is recommended to provide a reasonable explanation.
2) In the abstract, what does “corrA” mean?
3) In line 113, Eq.(11) should be Eq.(10), and one similar case occurs in line 197.
4) In section 5.1, please explain the type of the random distribution for APs and UEs, and offer more test parameters.
5) In Section 5.2, please explain the reason for the inconsistent orders of magnitude of the total downlink rate between Figure 2 and Figure 3. For instance, in Figure 3, around L=64 (M=30, K=20), the total downlink rates for the curves "optimal phase shift," "without optimization," and "optimal phase shift and location" are approximately 1.75, 1.15, and 0.25 bps/Hz, respectively, whereas in Figure 2, for M=30 (L=64, K=20), the total downlink rate is approximately 10 bps/Hz. The orders of magnitude in Figure 2 and Figure 3 are not consistent.
Comments on the Quality of English LanguageThis paper has good readability, but needs to be modified.
Reviewer 2 Report
Comments and Suggestions for Authors
This paper offers optimisation for Cell-Free mMIMO system with new UAV-RIS technology. The results seems interesting, because Authors take into account multiple UEs and their locations to derive closed form approximation for sum rate. But some improvements can be recommended.
Title doesn't reflect downlink scenario considered in the Article. Some phrases in the Abstract should be described more clearly: "corrA closed-form approximation", "optimal results".
Key-words should be added/corrected, e.g. downlink sum rate optimisation, Location and Trajectory Optimization, closed form approximation.
The introduction is well written, but it contains only one publication dated 2023 and two publications dated 2022; more recent publications need to be added. Also the area of application seems not to be clear: wireless networks, mobile networks, OMA/NOMA systems?
Area and position coordinates for UAV-RIS in system model should be described more clearly with the reference to Fig. 1.
Please check Rayleigh channel coefficient and complex value notation in (1) and other equations. Additive white Gaussian noise is also complex?
In Section 4 (Optimization) some reference for optimization methods used should be added.
In Section 5 location and trajectory simulation and selection should be described in more detail with the reference to Fig. 4 and Fig. 5. For Section 5.2 maybe additional table with simulation parameters is needed (K, L, N, etc.)
In the Conclusion section some conclusions about the proposed Theorem 1, its restrictions, assumptions and simulation accuracy should be added, e.g. the restriction of ideal CSI knowledge, and recommendations for application.
Comments on the Quality of English LanguageMisprints like "corrA' (line 4), "syatem" (line 71) should be corrected throughout the text. More detailed description of channel model notations in Sections 2 and 3, with references, like in Section 5.
